# Optimal performance of stand-alone hybrid microgrid systems based on integrated techno-economic-environmental energy management strategy using the grey wolf optimizer

**Ahmed Sahib Tukkee**[ID][1,2☯]*, **Noor Izzri bin Abdul Wahab**[ID][1☯], **Nashiren Farzilah binti Mailah**[ID][1‡], **Mohd Khair Bin Hassan**[1‡]

**1** Advanced Lightning Power and Energy Research (ALPER), Department of Electrical and Electronic Engineering, Faculty of Engineering, University Putra Malaysia (UPM), Serdang, Malaysia, **2** Department of Construction and Projects, University of Kerbala, Karbala, Iraq

☯ These authors contributed equally to this work.
‡ NFM and MKBH also contributed equally to this work.
* GS59655@student.upm.edu.my

## Abstract

Recently, global interest in organizing the functioning of renewable energy resources (RES) through microgrids (MG) has developed, as a unique approach to tackle technical, economic, and environmental difficulties. This study proposes implementing a developed Distributable Resource Management strategy (DRMS) in hybrid Microgrid systems to reduce total net percent cost (*TNPC*), energy loss ($P_{loss}$), and gas emissions (*GEM*) while taking the cost-benefit index (*CBI*) and loss of power supply probability (*LPSP*) as operational constraints. Grey Wolf Optimizer (GWO) was utilized to find the optimal size of the hybrid Microgrid components and calculate the multi-objective function with and without the proposed management method. In addition, a detailed sensitivity analysis of numerous economic and technological parameters was performed to assess system performance. The proposed strategy reduced the system's total net present cost, power loss, and emissions by (1.06%), (8.69%), and (17.19%), respectively compared to normal operation. Firefly Algorithm (FA) and Particle Swarm Optimization (PSO) techniques were used to verify the results. This study gives a more detailed plan for evaluating the effectiveness of hybrid Microgrid systems from a technical, economic, and environmental perspective.

## 1 Introduction

Electric power is the mainstay of economic growth and long-term infrastructure development in any country. With an ever-increasing population and major technological development, the electricity demand is continually rising, while existing energy sources are decreasing at alarming rates [1]. The electricity sector relies on fossil fuels by 85% to meet the global electricity demand and thus contributes significantly to the increase in harmful emissions and global

**Data Availability Statement:** All relevant data are within the manuscript and its Supporting information files.

**Funding:** The author(s) received no specific funding for this work.

**Competing interests:** The authors have declared that no competing interests exist.

warming [2]. Environmental concerns and limited fossil fuel resources prompted the electricity sector to upgrade the current energy systems and increase reliance on renewable energy resources (RES) [3]. However, the fluctuating of RES generation and its dependence on weather has added new challenges [4]. No single source of RES can meet the load requirements, so it becomes necessary to combine two or more different non-distributable resources (NDR) like RES and distributable resources (DR) such as micro-turbines (MT) in addition to energy storage systems (ESS) to form a hybrid Microgrid system (HMGs) [5]. RES are cheap, environmentally friendly, and sustainable energy sources while MT and ESS provide suitable backup power sources with stable electrical output and smoother power supply [6]. The optimum performance of HMGs is accomplished by determining the optimal sizes of the various energy resources and managing the various energy resources effectively in proportion to the economic, technological, and environmental impacts [7]. Particularly, an effective management plan and an optimized HMGs design can prevent unjustified rises in investment costs and protect the environment to enable the best trade-offs between the design objectives.

## 1.1 Literature review

Microgrids are a reliable and autonomous way to satisfy load requirements and enhance the security, quality, and dependability of the power supply. In previous studies, numerous optimization methodologies were implemented to identify the optimal performance of HMGs. Computational methods, as sophisticated mathematical optimization methodologies, have been used to handle optimization problems such as mixed-integer linear programming (MILP), mixed-integer quadratic programming (MIQP), non-linear programming (NLP), and dynamic programming (DP). The majority of these mathematical have significant shortcomings such as high computation durations and variable parameter determinations [8]and [9]. Researchers have recently turned to metaheuristics optimization strategies to improve the performance of HMGs due to their ability to solve optimization problems and overcome the shortcomings of previous techniques [10]. The interests of the researchers covered a wide range of economic, environmental, and reliability issues to enhance the performance of HMGs. From an economic aspect, the improved shuffled frog leaping algorithm (ISFLA) was employed in [11] to reduce energy costs and pollution of HMGs utilizing combined heat and power (CHP). In [12], the performance of a hybrid grid system was analyzed and various control strategies were developed to improve system performance by applying the Backward Search Algorithm (BSA) to overcome the problems of fuel prices and gas emissions. The authors of [13] used nine distinct intelligent strategies to determine the optimal levelized cost of energy and annual levelized cost of HMGs containing PV/biomass and three alternative energy storage unit technologies. Four distinct intelligent technologies were employed in [14] to minimize the cost of energy and loss of power supply potential (LPSP) of HMGs formed of PV /WT and energy storage systems. In [15], the multi-objective function is devised to calculate the cost of energy and gas emissions for a PV/WT/ESS-based hydroponic pump system (PHS) operating in grid-connected mode. In [16] mixed linear programming (MILP) technique was applied to improve the economic feasibility of HMGs consisting of PV, WT, an energy storage system, and a turbine engine. The main objective was to reduce the cost of energy and dependence on conventional resources. A multi-objective function was presented in [17] to determine the optimal size of an HMG consisting of PV, WT, and diesel generator and battery storage. To improve the environment and reduce emissions, [18] employed a genetic algorithm to reduce overall operational costs and carbon emissions. The influence of optimal energy storage sizes and their properties on carbon removal was explored. In [19], a hybrid system consisting of PV, WT, a diesel generator, and an energy storage system was

studied to determine the technical and environmental advantages and determine the best performance. The effects of using different quantities of RES on the system's economic and environmental performance have been studied in a variety of case studies. Limited opportunity programming was employed in the [20] to arrange the subsequent day for numerous microgrids. The system, which consists of both conventional and renewable energy generation units, operates in an uncertain environment. Using a customized version of the IEEE 33-bus test system, the methodology was validated. To minimize losses, improve the voltage profile, and increase the dependability of the microgrids, an optimal allocation of the combined heat and power system was presented in [21]. In [22], a Mixed Linear Programming (MILP) problem-based power management system was presented to investigate the operating behavior of MGs and reduce power losses. To identify the optimum size of MG a, a two-stage MG scheduling mechanism is suggested based on the group search optimizer (GSO) algorithm in [23]. The system stability index (SSI) is suggested to choose the MG's location in the IEEE 33-bus distribution systems. The optimal size of a PV/WT/Biomass-based system with an energy storage system is examined in [24] using the generalized reduced gradient algorithm for maximizing the demand-supply fraction and reducing power losses. In [25], a multi-objective crow search technique is introduced for designing PV/FC/Diesel-based HMGs with net present cost and losses of power supply probability as objective functions. According to them, the optimized HMGs are a cost-effective and dependable power source for distant region applications. Table 1 displays a summary of the literature review.

Most of the previous research was successful in improving HMG performance with enhanced convergence precision; nonetheless, study gaps remain. Two major issues that require additional research can be summarized.

First, finding the optimal performance of HMGs should take into consideration many economic, technological, and environmental concerns. Most prior research's objectives were

**Table 1. An overview of Microgrid system literature reviews.**

| Ref. | Year | Hybrid MG configuration | Optimization method | Objective function | Research limitations |
|------|------|-------------------------|---------------------|--------------------|--------------------|
| [26] | 2021 | WT/Diesel generator/ battery | HOMER | NPV | The obtained results were not validated or compared to other optimization techniques, and environmental factors were not included in the study. |
| [27] | 2020 | WT/diesel generator/ battery | iHOGA | COE | The hybrid system does not incorporate solar energy, nor are regional renewable resources well utilized. |
| [28] | 2020 | Biogas/Biomass/ PV/ WT/Fuel Cell | GA | COE | There haven't been any discoveries or advancements in optimization algorithms to better choose the hybrid energy system's ideal size. |
| [29] | 2020 | PV/WT/FC | SOA | COE, LPSP | The description of the study's findings is insufficient, and its consequences are not further explained. |
| [30] | 2022 | PV/WT/DG/BAT El | IGWO | COE | Lack of a more thorough cost-benefit analysis of hybrid systems and the hybrid system's pairing with only one energy storage |
| [31] | 2022 | PV/WT/Diesel/ battery | HSA | The annual cost of the system (ACS) | The proposed hybrid system's local contribution in terms of economy and reliability is not further discussed; only the performance of the optimization algorithm is compared. |
| [32] | 2020 | PV/WT/Diesel/ bat | PSO | TNPC | The effectiveness and reliability of the optimization algorithm in determining the optimal sizing are not compared or examined. |
| [33] | 2021 | PV/WT/BAT | IWOA | The weighted sum of the three costs | There is no more information available about hybrid systems, nor is their cost-effectiveness evaluated. |
| [34] | 2021 | PV/Wind/Diesel/ Battery | EO | NPC | More renewable components must be taken into consideration as well as more investigation of the optimization process. |
| [35] | 2022 | PV/Wind/Diesel/ Battery | IAOA | NPC | The implications of MG systems under various technological and economic circumstances are not examined. |

single functions that dealt with specific system parameters. However, it is critical to highlight and fully utilize the multi-objectives feature depending on distinct research objectives.

Second, most previous studies focused on the effect of fluctuations in RES generation and load disturbances on HMGs reliability, whereas the effect of employing efficient management strategies to manage the operation of distributable resources was not addressed. One of the most essential aspects is investigating and understanding how HMGs perform optimally in the context of distributable and RES to provide a stable local energy source.

## 1.2 Contributions of the study

This study contributed to resolving research gaps highlighted through the following contributions and innovation:

(1) A practical design framework for standalone HMGs incorporating PV, WT, micro-turbines (MT), and energy storage systems (ESS) was suggested. The cost of energy, energy losses, and gas emissions were examined independently as a single objective function then, the impact of all objectives was taken as a multi-objective function. The distributable resources management strategy (DRMS) was implemented to improve the HMGs' performance by reducing the cost of fuel consumption and harmful emissions.

(2) A further investigation is conducted on the number of major techno-economic indicator improvements. The sensitivity analysis outcomes can offer appropriate investment evaluation for the concerned authorities and investment operators. (3) Three innovative meta heuristic techniques GWO, FA, and PSO are implemented to solve the design problem and enhance the effectiveness of HMGs.

The rest of the paper is structured as follows. The mathematical formulation of HMGs are presented in Section 2. The proposed strategy, objective functions formulation, and operating constraints are presented in Section 3. The GWO optimization is presented in Section 4. The results are explored and discussed in Section 5. Finally, conclusions and future works are given in Section 6.

## 2 Architecture and modeling of HMGs

In this section, the architecture of the hybrid Microgrid system is presented, followed by mathematical models of all major components.

### 2.1 System architecture

The overall architecture and statistics for HMGs were taken from the standard test system [36]. The test system consists of 6 buses and 11 transmission lines. As shown in Fig 1, two separate RES of photovoltaic panels (PV) and wind turbines (WT) are connected to buses 2, 4, and 6, respectively. Bus 5 is supported by the energy storage system (ESS). The numbers on the transmission lines indicate the length and resistance of the line. These HMGs also feature distributable resources such as Micro-turbines (MT) connected to bus 1. S1 Table shows transmission line data for the distribution network.

### 2.2 Hybrid microgrid modeling

Accurate mathematical expression-based modeling of all MG components is necessary for analyzing while maintaining apprised system performance. The employed mathematical expressions are displayed as follows.

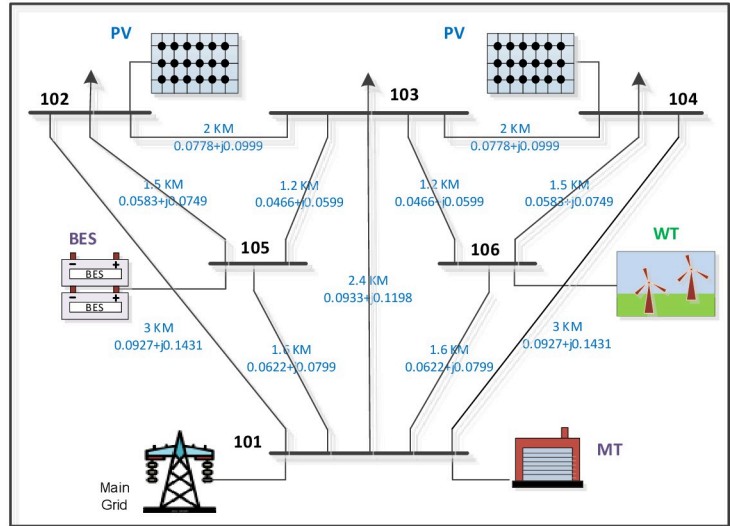

**Fig 1. Schematic diagram of a conceptual hybrid Microgrid system.**

**2.2.1 PV panels.** The output energy of PV panels at the time (t) ($E_{PV}(t)$) in [kW], can be estimated using(1) and (2) [37].

$$E_{PV}(t) = N_{PV}\left[R_{PV}f\left(PV\left(\frac{G(t)}{G_{ref}}\right)(1 + \propto_p(T_c(t) - T_{ref}))\right)\right] \times (1 - \mu) \tag{1}$$

$$T_c(t) = T_{amb}(t) + \left[\frac{(NOCT) - 20}{800} \times G(t)\right] \tag{2}$$

Where $N_{PV}$ is the number of PV panels, $R_{PV}$ is the PV-rated capacity[kW], $f_{PV}$ is the PV panels derating factor [%], $G(t)$ is the solar radiation [W/m2], $G_{ref}$ is the standard episode solar radiation [1000 W/m2], $\propto_P$ is the coefficient of temperature [%/°C], $T_c(t)$ is the cell temperature [°C], $T_{ref}$ is the standard cell temperature at test conditions [25 ∘C], $T_{amb}$ is the ambient temperature, $NOCT$ is the nominal operating temperature of a PV panel, and $\mu$ is the generative disturbance coefficient.

**2.2.2 Wind turbines.** Since the speed of the wind is very intermittent, the output energy of a wind turbine $E_{WT}(t)$ in [kW] is a function of wind speeds and is defined by (3) [38].

$$E_{WT}(t) = N_W T \begin{cases} 0 & v(t) \leq v_{cutin} or v(t) \geq v_{cutout} \\ P_r\left(\frac{v^3(t)-v^3_{cutin}}{v^3_r-v^3_{cutin}}\right) \times (1 - \mu) & v_{cutin} < v(t) < v_r \\ P_r(1 - \mu) & v_{cutin} < v(t) < v_r \end{cases} \tag{3}$$

Where, $N_{WT}$ is the number of WT, $v(t)$, $v_r$, $v_{cutin}$, $v_{cutout}$, and $P_r$ is the wind speed [m/s], rated speed [m/s], cut-in speed [m/s], cut-out speed [m/s], and output power at the rated speed [kW], respectively.

**2.2.3 Micro-turbines.** As a backup power supply, an MT generator is required. The influence of ambient temperature on MT energy output $E_{MT}(t)$ in [kW] is depicted in (4).

$$E_{MT}(t) = N_{MT}(-0.0113 \times T^2_{amb}(t) + 0.153 \times T_{amb}(t) + P_{MT,reat}) \tag{4}$$

Where $N_{MT}$ is the number of MT, and $P_{MT,reat}$ is the rated power of MT in [kW].

**2.2.4 Energy storage systems.** The incorporation of ESS improves system reliability and provides extra energy when renewable (PV/WT) are unavailable in HMGs. The charged energy of the ESS in [kW] can be calculated according to (5) [39].

$$E_{ESS}(t+1) = N_{ESS}(E_{ESS}(t) \times (1 - \sigma) + \left[ (E_{PV}(t) + E_{WT}(t)) - \frac{E_{load}(t)}{\eta_{inv}} \right] \times \eta_{ESS}) \qquad (5)$$

Where, $N_{ESS}$ refers to the number of ESS, $E_{ESS}(t+1)$ and $E_{ESS}(t)$ in [kW] are the charge quantities of ESS at the time $(t+1)$ and $(t)$ respectively, $\sigma$ is the self-discharge rate [%], $\eta_{inv}$ is the efficiency of the inverter, $E_{Load}(t)$ and $\eta_{ESS}$ are the load demand and charge efficiency of the ESS, respectively. However, the discharging energy of the ESS at time t can be calculated as in (6).

$$E_{ESS}(t+1) = N_{ESS}(E_{ESS}(t) \times (1 - \sigma) - \left[ \frac{E_{load}(t)}{\eta_{inv}} - (E_{PV}(t) + E_{WT}(t)) \right] \times \eta_{ESS}) \qquad (6)$$

# 3 Formulation of objective functions and constraints

The formulation of the objectives function and the operational constraints of the system are briefly explained in this section.

## 3.1 Objectives function formulation

The study issue is resolved by calculating the minimal value of three objective functions: total net present value cost (*TNPC*), power losses ($P_{loss}$), and gas emissions (*GEM*) while taking into consideration various equality and inequality restrictions. Eq (7) illustrates the general formulation of the research problems.

$$minimize(OF) = min(OF_1 + OF_2 + OF_3) \qquad (7)$$

Where, $OF_1$, $OF_2$, and $OF_3$ represent the objective functions; *TNPC*, $P_{loss}$, and *GEM* are to be minimized, respectively. The problematic triple-objectives function model can be transformed into a simple single-objective function by placing weighting parameters as shown in (8):

$$OF = min(w_1.TNPC + w_2.P_{loss} + w_3.GEM) \qquad (8)$$

Where $w_1$, $w_2$, and $w_3$ are the weighting variables for total energy costs, power losses, and emissions. The sum of the weight should equal 1.

**3.1.1 Objective 1.** The minimization of *TNPC* of *HMGs* is the main objective of the optimization problem. The *TNPC* is the sum of the capital investment cost ($C_c$), operation and maintenance cost ($C_{o\&m}$), replacement cost ($C_r$), fuel cost ($C_f$), and salvage cost ($C_s$) as in (9) to (16) [40, 41].

$$TNPC = C_c + C_{o\&m} + C_r + C_f - C_s \qquad (9)$$

$$C_c = (C_{c,PV} + C_{c,WT} + C_{c,MT} + C_{c,ESS} + C_{c,inv}) \qquad (10)$$

$$C_{o\&m} = (C_{o\&m,PV} + C_{o\&m,WT} + C_{o\&m,MT} + C_{o\&m,ESS} + C_{o\&m,inv}) \qquad (11)$$

$$C_r = \xi(C_{r,PV} + C_{r,WT} + C_{r,MT} + C_{r,ESS} + C_{r,inv}) \tag{12}$$

$$C_f = (Fuel_{MT} \times \rho_{fuel}) \tag{13}$$

$$C_s = PVF(C_{s,PV} + C_{s,WT} + C_{s,MT} + C_{s,ESS} + C_{s,inv}) \tag{14}$$

$$PVF = \frac{(1 + i_r)^n}{i_r(1 + i_r)^n} \tag{15}$$

$$\xi = \sum_n^n l \frac{1}{(1 + i_r)^n l} \tag{16}$$

Levelized cost of energy (LCOE) can be determined as in (17).

$$LCOE = \frac{TNPN}{\sum_{t=1}^{8760} E_{toteleneryserved}} \times CRF \tag{17}$$

CRF is the capital recovery factor as in (18).

$$CRF = \frac{1}{PVF} \tag{18}$$

Where $PVF$ is the present value factor, $\xi$ is the coefficient of a lump-sum payment, $i_r$ is the interesting factor, $n$ is the project lifetime, and $nl$ is the component lifetime.

**3.1.2 Objective 2.** The second objective of this study is to minimize the power losses of HMGs. Power losses are affected by the configuration of the MG, the voltage level of the buses, the permissible load levels at each bus, the length and size of the conductors, and the time-varying load profile [42]. Power losses $P_{loss}$ in [kW] are calculated using (19).

$$P_{loss(i,j)} = \left(\frac{P_{(i,j)}(t)}{V_i(t)}\right)^2 * \left(\frac{\rho}{A}\right) * C_l \tag{19}$$

Where, $P_{(i, j)}$ is the power transmission between bus $i$ and $j$ at time (t). $V_i$ is the distribution voltage at the ith bus, $\rho$ is the resistivity of the conductor, $A$ is the cross-section of the conductor, and $C_l$ is the length of the conductor connecting two buses. Thus the total $P_{loss}$ of the HMGs at any time can be calculated as in (20).

$$P_{loss}(t) = \sum_i^j \sum_{jwherei \neq j}^i P_{loss(i,j)}(t) \tag{20}$$

**3.1.3 Objective 3.** Environmental sustainability can be improved by reducing the amount of harmful gas emissions, especially carbon dioxide ($CO_2$). Gas emissions $GEM$ in [t/year] is considered as the third objective. The $CO_2$ weight is calculated using (21) ([43]).

$$CO_2 = (\frac{C_c}{E_{MT}})/1016.04 \tag{21}$$

Where, $C_c$ is the carbon element's proportion, which equals 0.6078 (kg/kWh). The total

emissions *GEM* (t/y) can be estimated as given in (22).

$$GEM = \sum_{i=1}^{NMT} CO_{2,MGi} \tag{22}$$

## 3.2 Constraints considered for the study

The research problem of this study was formulated according to two main types of constraints, one is the reliability constraint, and the other is the quantitative constraint of the MG components.

**3.2.1 Reliability constraints.**   Loss of power supply probability (LPSP). As a reliability indicator, LPSP is suggested, and it has a range of 0 to 1. The definition of LPSP is given following [44] as in (23) and (24):

$$LPSP = \frac{\sum_{t=1}^{T} LOE(t)}{\sum_{t=1}^{T} E_{load}(t)\triangle t} \tag{23}$$

$$LOE(t) = \frac{E_{load}(t)}{\eta_{inv}} - [E_{RES}(t)\triangle(t) + E_{ESS}(t) - E_{ESS,min}] \tag{24}$$

Cost-benefit index (CBI). The final benefits of the project can be calculated using the cost-benefit analysis index. The formula of the *CBI* is given in (25) [45]:

$$CBI = \frac{1 - LPSP}{TNPC} \times 100\% \tag{25}$$

**3.2.2 Quantitative constraint.**   The operational constraints of the component size are presented in (26) to (28).

$$E_{MT,min} \leq E_{MT} \leq E_{MT,max} \tag{26}$$

$$E_{PV,min} \leq E_{PV} \leq E_{PV,max} \tag{27}$$

$$E_{WT,min} \leq E_{WT} \leq E_{WT,max} \tag{28}$$

# 4 Management strategy and optimization technique

This section presents a brief overview of the proposed management strategy, DRMS, as well as the GWO method.

## 4.1 Distributable resource management strategy

The management of HMGs, which combine distributable and RES resources, is the most challenging issue for energy operators. In normal operating strategies, the net energy at each period is determined by comparing the amount of energy provided by RES and the load demand. If the net energy is greater than zero, it means there is a surplus of energy that can be used to charge the ESS until it reaches full capacity, and the rest is described as surplus energy. However, if there is an energy shortage i.e. the net energy is less than zero, the ESS is

used to compensate for the energy shortage until reaches the minimum level, then MT is used automatically to recompense the energy shortage and supply the loads [46]. In the proposed DRMS, the operation of MT is managed to reduce fuel consumption and emissions. When there is an enery shortage and there is not enough energy in the ESS, the MT is not started directly but the minimum load ratio of the MT is checked. The minimum generating load ratio is the ratio of the energy produced by the MT to compensate the load to the maximum generating capacity (before this value the fuel consumption rate is much higher and is frequently set at 30% of the maximum generating value) [47]. This value indicates that MT units start generating power only when the power shortage exceeds 30% of their rated capacity. Meanwhile, with the DRMS, the MT operates according to load requirements and the price of electrical in the market. After the shortage energy requirements are met, the extra energy from MT is used to charge the ESS. As a result, the consumption of fuel and ESS discharges is reduced continuously. The general layout of the suggested strategy is depicted in Fig 2.

## 4.2 The proposed optimization technique

The Grey wolf optimizer GWO is utilized to determine the optimal objectives function before and after the DRMS method is implemented. The GWO is introduced to emulate the social structure and predating of grey wolves, with streamlined operation, fewer parameter modifications, and understandable and obvious principles [48]. In the GWO approach, the wolves were divided into four levels: level 1 is the present best individual

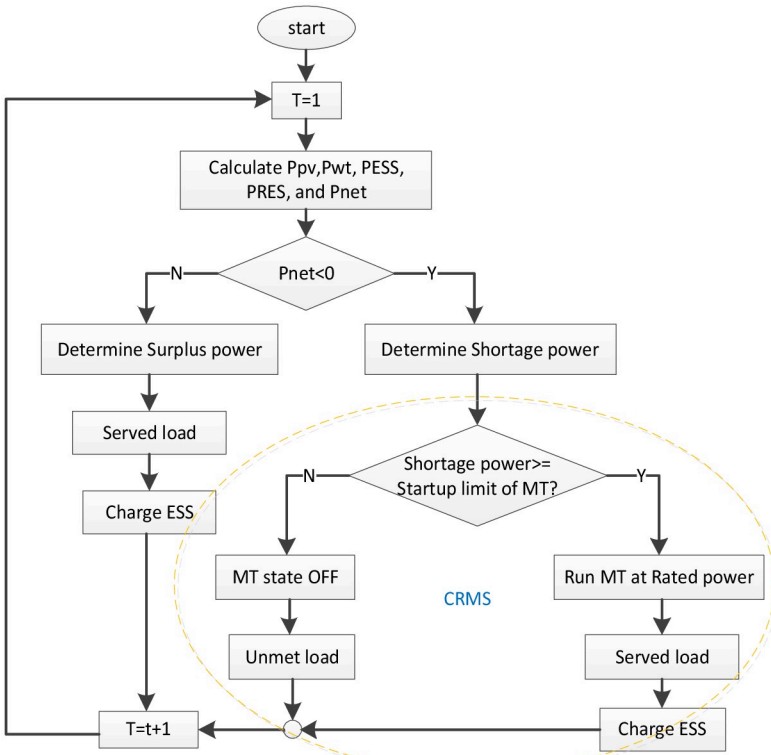

**Fig 2. Illustrates the general layout of the DRMS approach.**

wolf (leader), established by the letter $\propto$, levels 2 and 3 reflect the sub optimal second and third-best alternatives and indicated by the letters $\beta$ and $\delta$, and level 4 corresponds to the ordinary alternative, represented by the letters $\sigma$. The main phases of the GWO strategy are surrounding, hunting, and attacking prey. Grey wolves follow the following strategies to achieve the goal: Surrounding: The prey is surrounded by GWs from all sides, as seen in (29) and (30).

$$\vec{D} = |\vec{C} \times \vec{X}_p - \vec{X}(t) \tag{29}$$

$$\vec{X}(t+1) = \vec{P}(t) - \vec{A} \times \vec{D} \tag{30}$$

Where, $\vec{X}(t)$, and $\vec{X}_p(t)$ is the Wolfe and prey position vectors respectively, and $\vec{A}$, $\vec{C}$ represent the vector evaluated in (31).

$$\vec{A} = 2ar_1, \vec{C} = 2r_2 \tag{31}$$

Where $r_1$, and $r_2$ are random number ranges from 0—1. The components of $a$ linearly decrease from 2 to 0 over time as in (32).

$$a = 2 - \frac{iteration}{max.iteration} \tag{32}$$

Hunting: In a hunting activity, $\propto, \beta$, and $\delta$ are thought to have a clearer view of the prey's location. The other wolves $\sigma$ are forced to follow the leader wolves to find prey. This hunting activity is modeled by (33).

$$\vec{D}_\propto = |\vec{C}_1 \vec{X}_\propto - \vec{X}(t)|, \quad \vec{D}_\beta |\vec{C}_2 \vec{X}_\beta - \vec{X}(t)|, \quad \vec{D}_\delta |\vec{C}_3 \vec{X}_\delta - \vec{X}(t)| \tag{33}$$

Where $\vec{X}_\propto$, $\vec{X}_\beta$, and $\vec{X}_\delta$ are the best three solutions at each iteration, $C_1$, $C_2$, and $C_3$ can be calculated as in (32). The new location of the three best wolves can be calculated (34).

$$\vec{X}_{i1} = \vec{X}_\propto - \vec{A}_1 \times \vec{D}_\propto, \quad \vec{X}_{i2} = \vec{X}_\beta - \vec{A}_2 \times \vec{D}_\beta, \quad \vec{X}_{i3} = \vec{X}_\delta - \vec{A}_3 \times \vec{D}_\delta \tag{34}$$

Then the new location of the best wolf is calculated according to the (35).

$$\vec{X}(t+1) = \frac{\vec{X}_{i1} + \vec{X}_{i2} + \vec{X}_{i3}}{3} \tag{35}$$

Attacking: The wolves begin attacking the prey after they stop moving and the hunt is over. Fig 3 illustrates an overall flowchart of the GWO optimizer.

## 5 Results and discussion

The effectiveness of CRMS on overall system performance was validated using four specified scenarios based on the intended objective function.

### 5.1 Results of hybrid MGs

Four different scenarios were addressed to evaluate the system's performance. *TNPC*, $P_{loss}$, and *GEM* are all considered separately as a single objective function when formulating the first, second, and third scenarios. In the fourth scenario, the weighted sum approach was employed as the main metric to attain numerous equivalent objectives. The weighted sum approach is one of the most utilized methods for transforming multiple optimization problems into single objective problems. The GWO, GA, and PSO techniques were used to optimize the

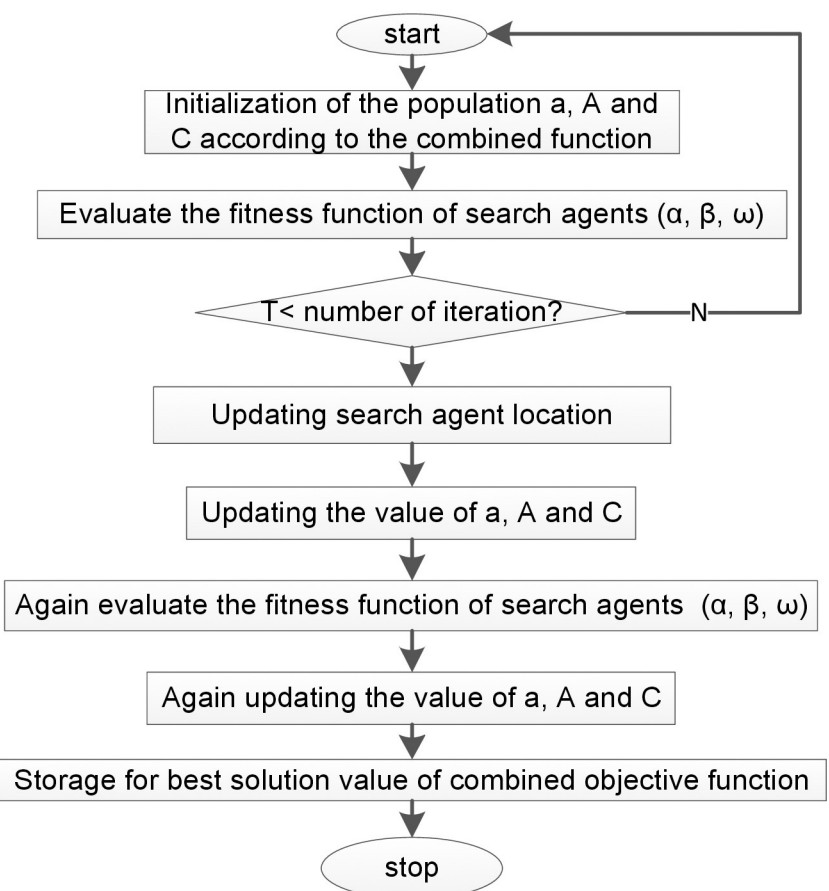

**Fig 3. The general outline of the GWO implementation steps to find the optimal size of the HMGs.**

performance of the HMGs. All techniques have the same optimization parameters: population size (200), and an iterative procedure limit (100). The period for data processing is 1 hour and the data length is 8760 h. MG project lifetime is assumed 20 years, and the standard rate of interest is 6%. Fig 4 displays climate and load demand data for the research area located in Kuala Lumpur, Malaysia (Latitude 2.9977 and Longitude 101.714), which are obtained from the National Aeronautics and Space Administration (NASA). S1 Fig shows annual weather data for the study area.

Tables 2 and 3 provide information on all technical and economic parameters of the MG components. The objective function of each population is calculated in each generation, and the population, that cannot meet the predetermined constraints, is discarded from the next generation to continue the process until the maximum generation is reached.

**5.1.1 First scenario.** The optimization objective in the first scenario is to reduce the TNPC of HMGs given reliability limitations. GWO, FA, and PSO were used to calculate the objective function. Table 4 shows the results of the three optimization methods in both cases with and without the use of DRMS. When the HMGs operating under DRMS, the optimal number of components were 173 PV panels, 64 WT, 2 MT, and 405 ESS, TNPC is 1.8412 $M\$$, and LCOE is 0.1264 $/kWh. While it reached 1.9041$M\$$ for TNPC, and 0.1281$/kWh of LCOE

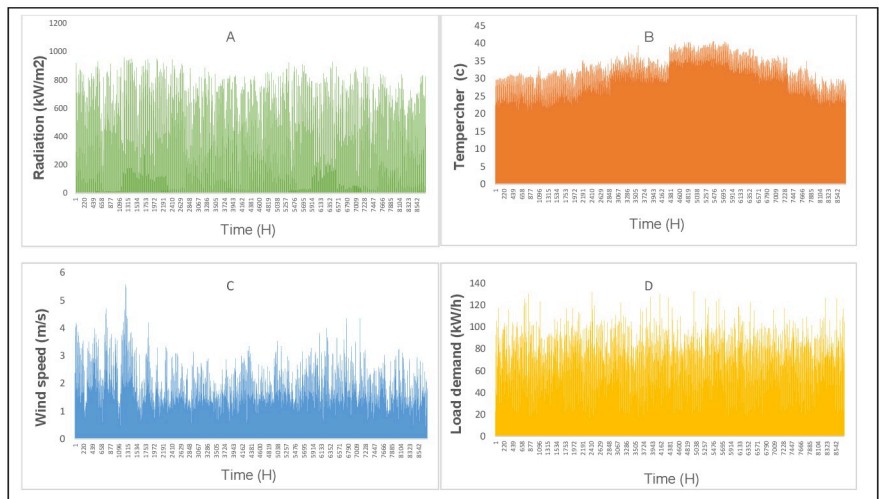

**Fig 4. The weather data and load demand for the study area for one year.** (A) Solar radiation, (B) temperature, (C) wind speed, and (D) annual load demand.

**Table 2. The MG technical parameters [49, 50].**

| Equipment | Parameter | Value | Equipment | Parameter | Value |
|---|---|---|---|---|---|
| PV | Rated capacity | 1k W | ESS | Nominal capacity | 1kWh |
| | Nominal temp. | 146˚C | | charge efficiency | 95% |
| | Efficiency at STC | 85% | | Discharge efficiency | 90% |
| | Derating factor | 13% | | Depth of discharge | 0.8 |
| WT | Rated capacity | 1k W | | Self-discharge rate | 2% |
| | Cut-in wind speed | 2.4 m/s | MT | Rated capacity | 0–25 kW |
| | Cut-out wind speed | 20 m/s | | Efficiency | 85 |
| | Rated wind speed | 11 m/s | Inverter | efficiency | 95% |

**Table 3. Economic parameters of MG [49], [50].**

| Equipment | Cc($/unit) | Co&m ($/unit/y) | Cr($/unit) | Cf ($/l) | Lifetime(y) |
|---|---|---|---|---|---|
| PV | 2000 | 20 | 1800 | 0 | 20 |
| WT | 3200 | 32 | 3000 | 0 | 20 |
| ESS | 750 | 8 | 700 | 0 | 10 |
| MT | 12000 | 300 | 10000 | 0.2 | 10 |
| Inverter | 800 | 7 | 750 | 0 | 15 |

when the system operates without DRMS. Fig 5 shows the LPSP and CBI for the two operating modes.

**5.1.2 Second scenario.** Minimizing power loss is the main objective in the second scenario. The outcomes of optimization utilizing various strategies are also shown in Table 5. It is obvious that the suggested GWO results in a minimum power loss of 1.5234 kW under DRMS after it was 1.6254 kW in normal operating with a discernible rise in TNPC and LCOE. Fig 6 shows the LPSP and CBI index of the system when power losses are adopted as a single objective function.

**Table 4. Optimization parameters for the HMGs in Scenario 1.**

| Variable | Normal operating | | | Operating with DRMS | | |
|---|---|---|---|---|---|---|
| | GWO | FA | PSO | GWO | FA | PSO |
| N.PV | 168 | 170 | 169 | 173 | 175 | 176 |
| N.WT | 65 | 65 | 67 | 64 | 63 | 63 |
| N, MT | 3 | 3 | 3 | 2 | 2 | 2 |
| N.ESS | 410 | 411 | 413 | 405 | 407 | 404 |
| $LCOE(\$/kWh)$ | 0.128 | 0.129 | 0.129 | 0.126 | 0.127 | 0.127 |
| **TNPC ($/y)** | 1.904 | 1.924 | 1.924 | 1.841 | 1.854 | 1.861 |
| $P_{loss}$ (kW) | 3.525 | 3.594 | 3.655 | 3.494 | 3.514 | 3.586 |
| Emission (t/y) | 0.065 | 0.065 | 0.066 | 0.061 | 0.062 | 0.062 |

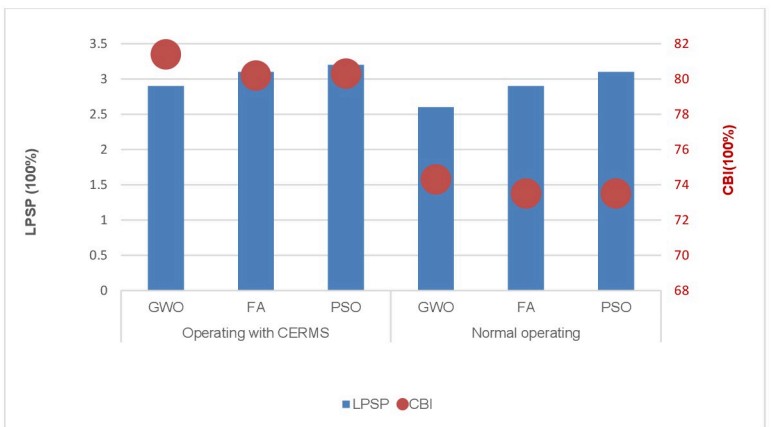

**Fig 5. LPSP and CBI indices of the first scenario under two operating modes.**

**5.1.3 Third scenario.** As the third objective set for the system, pollutant emissions are to be kept to a minimum. The amount of MT energy used to meet demand directly correlates with the total emissions. Since the environmental goal of the optimization is to lower the total GEM, the executed algorithm investigates the various sizes of the components by calculating the total GEM to ultimately arrive at the lowest value of emissions. The comparison between

**Table 5. Optimization parameters for the HMGs in Scenario 1.**

| Variable | Normal operating | | | Operating with DRMS | | |
|---|---|---|---|---|---|---|
| | GWO | FA | PSO | GWO | FA | PSO |
| N.PV | 171 | 173 | 179 | 172 | 174 | 178 |
| N.WT | 72 | 71 | 73 | 74 | 73 | 73 |
| N, MT | 3 | 3 | 3 | 3 | 3 | 3 |
| N.ESS | 421 | 419 | 420 | 424 | 426 | 425 |
| $LCOE(\$/kWh)$ | 0.132 | 0.134 | 0.135 | 1.030 | 0.131 | 0.132 |
| TNPC ($/y) | 1.984 | 1.984 | 1.987 | 1.917 | 1.926 | 1.932 |
| $P_{loss}(kW)$ | 1.625 | 1.749 | 1.894 | 1.523 | 1.632 | 1.648 |
| Emission (t/y) | 0.052 | 0.056 | 0.058 | 0.048 | 0.049 | 0.049 |

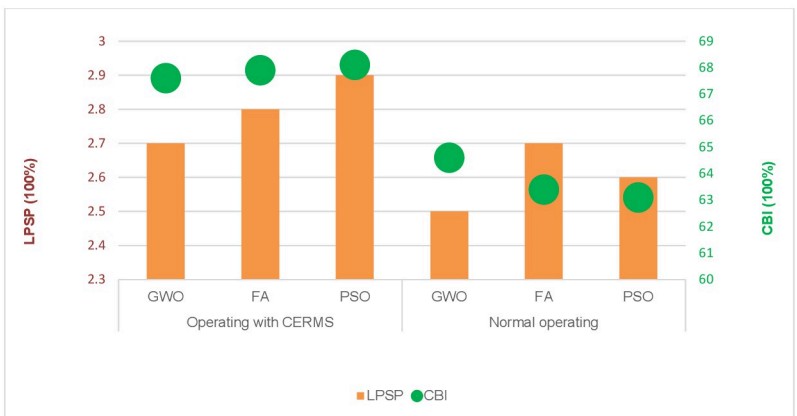

**Fig 6. LPSP and CBI indices of the second scenario under two operating modes.**

the optimizers under consideration with DRMS and normal operating is shown in Table 6. The suggested GWO achieves a minimal emissions level of 0.0112 t/y with a power loss of 2.4952 kW under DRMS while the emission level is 0.01621 t/y and power loss is 2.9547 kW in normal operation. Fig 7 depicts the LPSP index utilizing the suggested GWO and further optimizations under two operating modes.

**5.1.4 Fourth scenario.** The trade-off between the objectives makes the single objective function study a less desirable option as the number of engineering challenges rises. All parameters that affect system performance, even to varied degrees, must be considered for effective design by using multi-objective function techniques. Due to its simplicity and great effectiveness, the weighted summation approach is one of the most used techniques for addressing multi-objective problems. In the fourth scenario, the three objective functions were ranked in order of significance to the decision-maker; the energy cost is the primary priority of the MG operators, followed by power loss reduction, and finally, reducing gas emission levels. A set of appropriate weights for energy costs, power losses, and emission levels are calculated using GWO based on the significance of each objective. The potential weight sets within the specified range for each aim were constructed as a population matrix. The values of the weights were assumed to be positive and restricted where $W_1$ related to the cost of energy is restricted from 0.35 to 0.65, $W_2$ related to power losses is restricted from 0.20 to 0.50, and $W_3$ related to emission is restricted from 0.1 to 0.40. The weight set that achieves the least objective function is

**Table 6. Hybrid MG optimization parameters based on emission reduction.**

| Variable | Normal operating | | | Operating with DRMS | | |
|---|---|---|---|---|---|---|
| | GWO | FA | PSO | GWO | FA | PSO |
| N.PV | 183 | 186 | 187 | 193 | 197 | 198 |
| N.WT | 81 | 82 | 81 | 85 | 85 | 86 |
| N, MT | 2 | 2 | 2 | 1 | 1 | 1 |
| N.ESS | 476 | 485 | 482 | 488 | 486 | 489 |
| $LCOE(\$/kWh)$ | 0.133 | 0.133 | 0.134 | 0.131 | 0.131 | 0.131 |
| TNPC (\$/y) | 1.875 | 1.873 | 1.879 | 1.786 | 1.784 | 1.789 |
| $P_{loss}$ (kW) | 2.954 | 2.987 | 3.125 | 2.495 | 2.853 | 2.365 |
| **Emission (t/y)** | 0.016 | 0.016 | 0.019 | 0.011 | 0.012 | 0.012 |

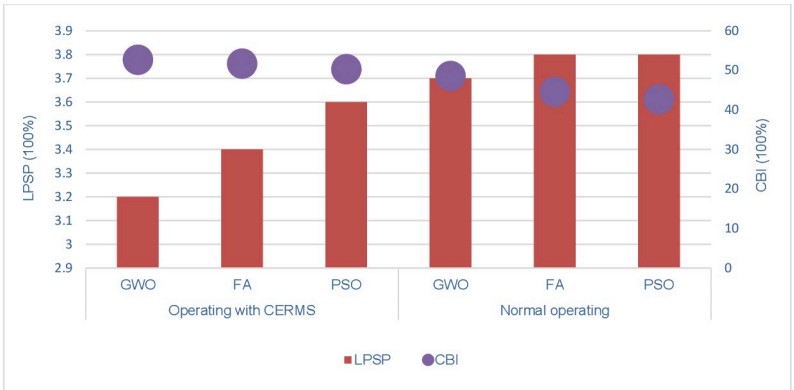

**Fig 7. LPSP and CBI indices of the third scenario under two operating modes.**

chosen as the multi-objective function weights which are 0.514, 0.297, and 0.189, for $W_1$, $W_2$, and $W_3$ respectively. Table 7 compares the variables obtained by GWO to those obtained by other optimizer. It is noted that the proposed DRMS has achieved an improvement in the HMGs indicators, where the LCOE was 0.127 $/kW, the TNPC was 1.8581 $/y, power losses were 2.2391 kW, and the emission level was 0.0183 t/y. Fig 8 illustrates the LPSP and CBI indices.

The fourth scenario's outcomes combine the system's operational, economic, and environmental objectives in a way that can ensure peak performance. Fig 9 compares the convergence of used AI techniques to find the optimal solution while the system operates with DRMS. The GWO achieves the fastest convergence rate when compared to FA and PSO since it has the specific advantage of maintaining high convergence accuracy and optimization problem tolerance with high nonlinearity and non-torsion tolerance. Also, the GWO-examined technique converges after approximately 28 iterations, whereas other methods' convergence curves saturate after about 40 rounds. The computation of the quantity of excess and shortfall energy for each period is an important parameter for the optimal performance of HMGs, as these values can be employed in the case of the system operating in grid-connected mode or as a multi-microgrid system. Fig 10 depicts the system's surplus and shortage energy levels for the two operating modes.

To provide an entire overview of the energy flow from the components of HMGs, Fig 11 demonstrates the output obtained using GWO for a typical week. The system's greater reliance

**Table 7. HMGs optimization parameters based on multi-objective function.**

| Variable | Normal operating | | | Operating with DRMS | | |
|---|---|---|---|---|---|---|
| | GWO | FA | PSO | GWO | FA | PSO |
| N.PV | 178 | 179 | 177 | 183 | 184 | 183 |
| N.WT | 77 | 76 | 78 | 79 | 80 | 81 |
| N, MT | 3 | 3 | 3 | 2 | 2 | 2 |
| N.ESS | 445 | 451 | 453 | 452 | 457 | 458 |
| $LCOE(\$/kWh)$ | 0.128 | 0.128 | 0.129 | 0.127 | 0.128 | 0.128 |
| TNPC ($/y) | 1.878 | 1.879 | 1.879 | 1.858 | 1.859 | 1.859 |
| $P_{loss}$ (kW) | 2.452 | 2.458 | 2.697 | 2.239 | 2.266 | 2.312 |
| **Emission (t/y)** | 0.022 | 0.026 | 0.028 | 0.018 | 0.020 | 0.021 |

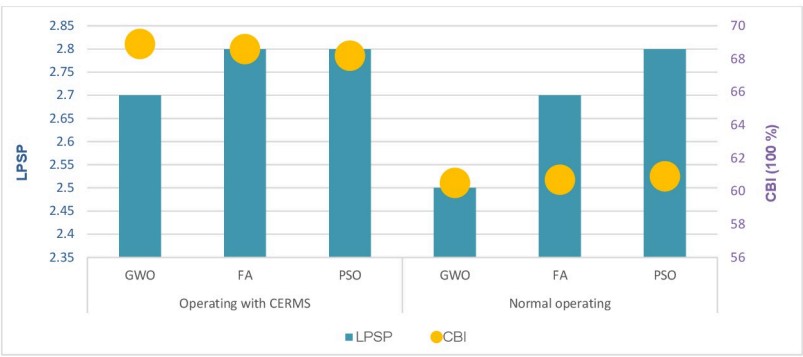

**Fig 8. LPSP and CBI indices of the fourth scenario under two operating modes.**

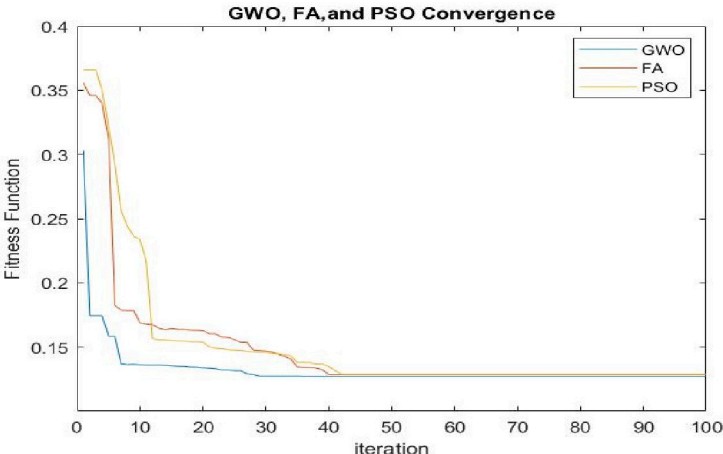

**Fig 9. Convergence of GWO, FA, and PSO techniques used to find the optimal solution.**

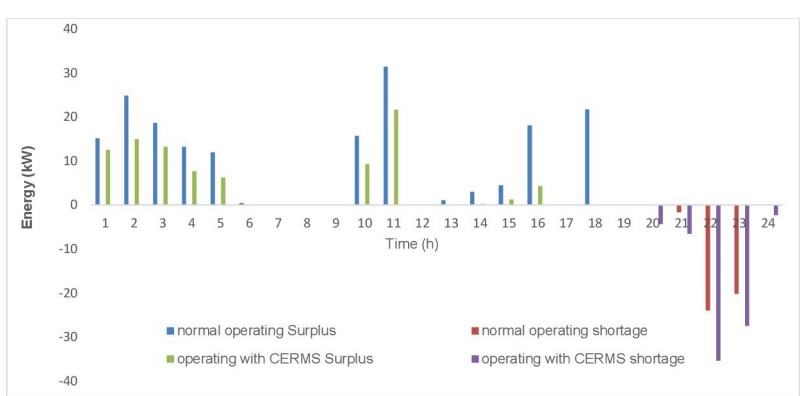

**Fig 10. Surpluses and shortage energy of HMGs in two operating modes.**

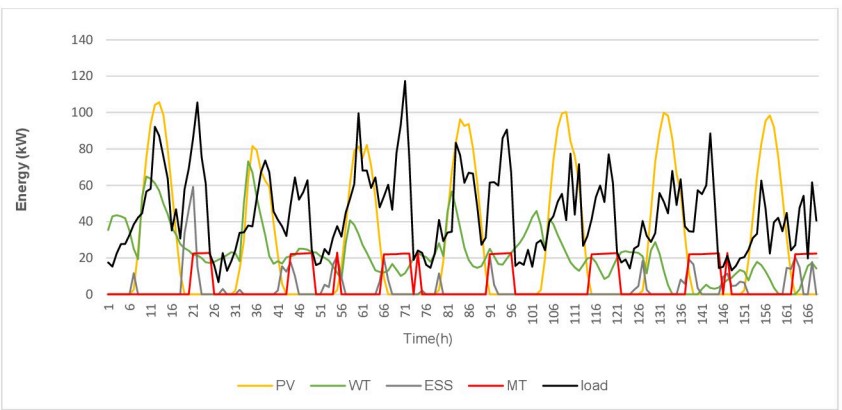

**Fig 11. The HMGs' hourly output for one week whilst operating under the DRMS, as determined by GWO.**

on RES and use of extra energy to charge the ESS are both effects of the DRMS implementation.

## 5.2 Sensitivity assessment analysis

The performance of the HMGs was investigated in this section along with several significant technical and economic factors. The sensitivity analysis approach depends on evaluating the effects of changing fundamental variables and uncertainties on the economic viability of an engineering project and determining whether the project's expenses can be comfortably borne when certain variables reach critical boundaries. Consequently, the HMGs' overall performance in many technical and economic circumstances is evaluated to provide reliable investment requirements for the implementation of more practical MGs. The operating startup limit of the MT, the interest rate, and the failure rate of renewable are the major variables evaluated in this study for system sensitivity analysis. A GWO was utilized to determine the repercussions of each difference to explain and assess the practical results of the investigation.

**5.2.1 MT startup limit variations.** Micro-turbines are used as backup power sources in HMGs to compensate for extreme power shortages. MT significantly contributes to rising emissions and rising operating costs for the system. As part of a planned management strategy, this study established an individual limit on the utilization of MT. According to the optimization results in Table 8, the MGs at 15%, 25%, 30% (baseline), and 35% MT startup limits demonstrate an increasing trend of TNPC and LPSP while decreasing in the GEM, indicating a drop in cost and power supply reliability.

The LCOE approaches 0.1272 $kWh$ with a 35% threshold and a GEM of 0.0098%. The same conclusion is drawn in normal usage, the total system cost reduces while the gas emission

**Table 8. Results of HMGs under different MT startup limit scenarios using GWO.**

| St.up(%) | Operating with DRMS | | | | Operating without DRMS | | | |
|---|---|---|---|---|---|---|---|---|
| | TNPC(M$) | LCOE ($/kWh) | LPSP | GEM (t/y) | TNPC(M$) | LCOE ($/kWh) | LPSP | GEM (t/y) |
| 15 | 1.768 | 0.126 | 0.025 | 0.082 | 1.802 | 0.127 | 0.022 | 0.098 |
| 25 | 1.798 | 0.126 | 0.026 | 0.047 | 1.837 | 0.128 | 0.024 | 0.062 |
| 30 | 1.858 | 0.127 | 0.027 | 0.018 | 1.878 | 0.128 | 0.025 | 0.022 |
| 35 | 1.932 | 0.127 | 0.029 | 0.009 | 1.921 | 0.128 | 0.028 | 0.011 |

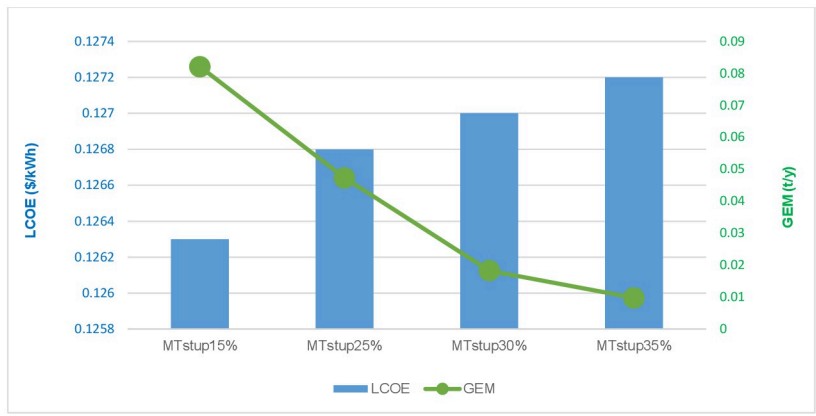

**Fig 12. Optimal indicators of the HMGs under different startup limits of MT using GWO.**

dependability rises as the MT startup limit rises, and vice versa. Reducing the MTI's starting limit extends its operating hours, which increases fuel costs and harmful emissions, and decreases the usage of RES, therefore, a larger hybrid system is needed to fulfill the necessary load supply. The inverse is also accurate. Fig 12 shows the optimal parameters of the HMGs under different startup limits of MT. The investigation into economic and environmental factors showed that the operating startup limits of the MT have broad implications for the overall benefit of the system, so they must be taken into consideration when designing HMGs.

**5.2.2 Interest rate variations.** Most governments are increasing their support for clean energy development to reduce emissions and achieve "carbon neutrality" by offering autonomous lending to the burgeoning clean energy sector. The impact of interest rate variations is very important to determine the cost-effectiveness of HMGs. Selecting an appropriate interest rate increases the economic returns on investments made in the clean energy industry, decreases unneeded investments, and promotes the expansion of clean hybrid MGs. The impact of altering the interest rate by 3%, 6% (baseline), 9%, and 12% while holding the other variables constant is covered in this paper. As indicated in Table 9, the TNPC value of the HMGs is increasing while the LCOE is decreasing, and the reliability of the power supply is deteriorating as the rate grows. In specific terms, the TNPC value under DRMS is 2.0541 M$, 1.8581 M$, 1.7523 M$, and 1.6531 M$ as the interest rate rises from 3% to 12% respectively. In addition, the system LCOE value increases from 0.1214 to 0.1321 and the system LPSP from 0.022 to 0.034 when the rate rises from 3% to 12%. Normal operational variations follow a trend that involves increasing TNPC values at 3%, 6% (baseline), 9%, and 12% interest rates, while LCOE decreases, and power supply reliability deteriorates.

**Table 9. Results of HMGs under different MT startup limit scenarios using GWO.**

| Ir(%) | Operating with DRMS | | | | Operating without DRMS | | | |
|---|---|---|---|---|---|---|---|---|
| | TNPC(M$) | LCOE ($/kWh) | LPSP | GEM (t/y) | TNPC(M$) | LCOE ($/kWh) | LPSP | GEM (t/y) |
| 3 | 2.054 | 0.121 | 0.022 | 65.24 | 2.143 | 0.127 | 0.021 | 58.41 |
| 6 | 1.858 | 0.127 | 0.027 | 69.32 | 1.876 | 0.128 | 0.025 | 60.04 |
| 9 | 1.752 | 0.130 | 0.031 | 70.41 | 1.778 | 0.123 | 0.029 | 62.36 |
| 12 | 1.653 | 0.132 | 0.034 | 71.35 | 1.725 | 0.138 | 0.031 | 63.52 |

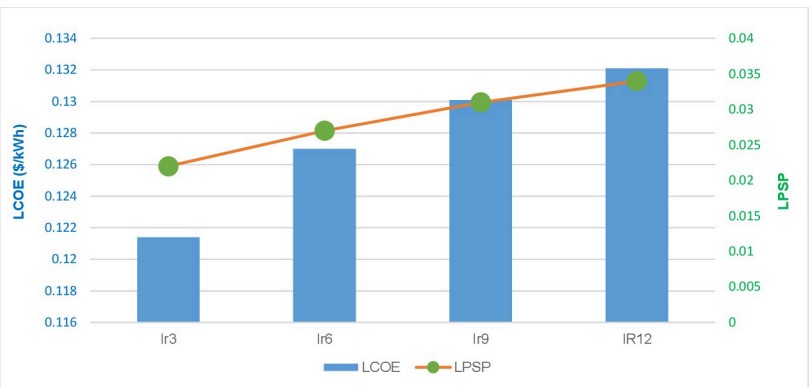

**Fig 13. Results of HMGs with DRMS under different interest rate scenarios using GWO.**

Fig 13 compares TNPC and BLPS when the hybrid system operates with DRMS under different interest rate scenarios. This figure shows that as interest rates climb more, the LCOE will continue to rise. However, the system's power supply will become less dependable. The analytical results demonstrate that an increase in the interest rate limits MG operators' ability to raise cost expenditures and decreases system reliability. In this situation, financial institutions should take an appropriate interest rate into account to guarantee investments and prevent irrational capital investment

**5.2.3 Generative disturbance coefficient variations.** The rate of generation of RES is affected by many different technical and atmospheric factors, which affect the reliability and estimation of the optimal size of the system components. The generative disturbance coefficient $\mu$(%) changes the rate at which energy is generated from RES. The disturbance factor must be maintained to a minimum to maximize the energy generated from RES. The optimization results are obtained using the generative disturbance coefficient in the ratios of 0% (baseline), 5%, 10%, and 15%. Fig 14 and Table 10 display the evaluation outcome.

Fig 14 demonstrates that the LCOE and LPSP are identically decreasing as the generative disturbance coefficient of RES decreases from 15% to 0%. Similarly, the CBI of hybrid MGs rises from 64.25% to 69.32% when the generative disturbance coefficient falls from 15% to 0. Recently, the generative disturbance coefficient notion has been steadily incorporated while

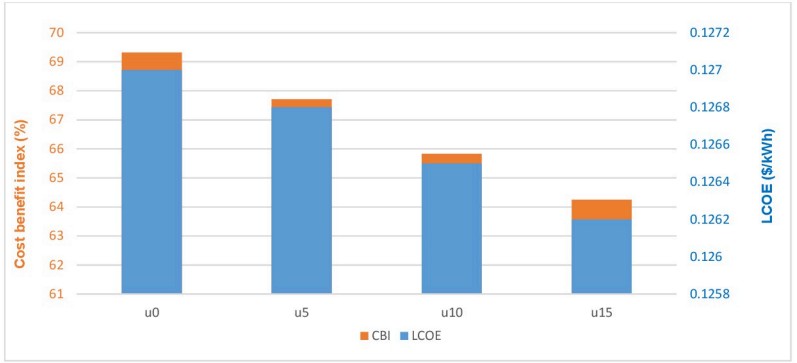

**Fig 14. Cost-benefit index and LPSP of the HMG system under different generative disturbance coefficients using GWO.**

**Table 10. Optimal parameters of HMGs under different disturbance coefficients using GWO.**

| $\mu$(%) | Operating with DRMS | | | | Operating without DRMS | | | |
|---|---|---|---|---|---|---|---|---|
| | TNPC(M$) | LCOE ($/kWh) | LPSP | GEM (t/y) | TNPC(M$) | LCOE ($/kWh) | LPSP | GEM (t/y) |
| 15 | 1.954 | 0.1262 | 0.033 | 64.25 | 1.978 | 0.1277 | 0.032 | 56.37 |
| 10 | 1.941 | 0.1265 | 0.031 | 65.83 | 1.948 | 0.1279 | 0.030 | 57.94 |
| 5 | 1.895 | 0.1268 | 0.029 | 67.71 | 1.898 | 0.1282 | 0.028 | 59.28 |
| 0 | 1.858 | 0.1270 | 0.027 | 69.32 | 1.876 | 0.1285 | 0.025 | 60.04 |

building renewable energy systems, although it has not been completely emphasized. The low generative disturbance coefficient of REs can improve the performance of MGs resulting in a dependable power source.

## 5.3 Results discussion

The main objective of this study is to determine the optimal size of HMGs components able to meet the electrical load demand with high reliability and efficiency. A rigorous strategy has been implemented to improve the penetration of RES, decrease fuel consumption, and reduce harmful emissions. The experimental results demonstrated that the GWO technology outperformed two other well-known algorithms, FA and PSO, to determine an optimal set of solutions and guarantee the reliability of the HMGs. When all economic, environmental, and operational factors are considered to determine optimal performance, as in the fourth scenario, the proposed DRMS strategy helped improve the TNPC, Ploss, and GEM values to 1.8581 M$, 2.2391 kW, and 0.0183 t/y, respectively, after it was 1.8781 M$, 2.4523 kW, and 0.0221 t/y in normal operation. The results of the optimization indicated that the suggested management method for HMGs provides higher economic, dependability, and environmental effectiveness, making it more attractive to consumers and energy investors. The impact of numerous technical and economic parameters, such as MT startup limit, interest rates, and generating disturbances, on the optimal design of the HMG was investigated. In terms of selecting the interest rate, the sensitivity analysis revealed that as the interest rate rises, the energy cost falls while the LPSP index rises, lowering the system's reliability. In other words, the costs and reliability of systems decrease in direct proportion to the rate of interest. Also suggests that setting the effective interest rate by stakeholders can influence the relationship between the design of renewable energy systems and their financial ramifications, enhance investment in the renewable energy sector, and contribute to avoiding irrational investments. In terms of reduction of energy cost and emissions, the decision on a start-up limit of MG is crucial due to the continuing high cost of fuel and emission restrictions. A sensitivity study of the results of this parameter showed that as the threshold limit is raised, the TNPC of the HMGs increases while the LPSP decreases. This means that as the operating startup limit of MT increases, the overall cost of HMGs increases. In addition, the emission rate rises as the startup decreases and vice versa. Here the investor can determine the appropriate operating startup limit in the most economical way based on the sensitivity analysis. Fluctuating RES generation can harm HMG performance. Higher failure probabilities might result from an increase in the TNPC and a drop in system reliability. Predicting proper disturbance coefficients for RES is therefore critical to avoid irrational capital spending and offer sufficient reliability.

## 6 Conclusions and future work

The purpose of this research is to determine the optimal performance of HMGs that incorporate RES and distributable energy sources. To meet the annual load supply with high dependability, the lowest energy cost, and the lowest emissions while taking operational and reliability restrictions like LPSP and CPI into account. To govern and more effectively monitor and confirm the HMGs' economic and environmental objectives, a developed technique known as DRMS was employed. A sensitivity analysis was also performed for the economic and operational parameters influencing the overall performance of the system. Based on the findings, the following conclusions can be established:

1) The suggested strategy successfully reduced TNPC and emissions while maintaining an acceptable LPSP rate across all artificial intelligence techniques. However, the GWO algorithm outperformed FA and PSO because of its low convergence and strong robustness.

2) The TNPC, LCOE, power losses, and emission levels of the HMGs with DRMS acquired by GWO are 1.8581 M\$, 0.1270 \$/kWh, 2.2391 kW, and 0.0183 t/y respectively, whereas it was 1.8781 M\$, 0.1285 \$/kWh, 2.2523 kW, and 0.0221 t/y respectively in normal operating.

3) The cost-benefit index increased from 61.72% to 68.31% when DRMS was implemented. Even though the LPSP was 0.027 a little higher than in the normal operating 0.025, this difference could be overlooked given the advantages for the economy and environment.

4) The investigation and analysis of various economic and technical parameters findings showed that an increase in MT startup limits and an increase in interest rates reduce the cost of energy, they also increase power loss rates and have an impact on the system's overall performance through generating disorders.

This study concentrates on developing optimum hybrid stand-alone MGs, but it can also be applied to grid-connected MGs for safer and more dependable energy supply in further works. Other diverse energy sources, such as geothermal, hydrogen cells, and so on, can also be used in the hybrid MGs and tested, which will be a challenge for future studies.

## Supporting information

**S1 Table. Microgrid data.**
(TIF)

**S1 Fig. Annual weather data.**
(TIF)

## Acknowledgments

The authors of this study express their sincere gratitude to the entire staff at the Advanced Lighting Power and Energy Research (ALPER), Department of Electrical and Electronic Engineering, Faculty of Engineering, University Putra Malaysia. Additionally, we extend our appreciation to the University of Karbala in Iraq.

## Author Contributions

**Conceptualization:** Ahmed Sahib Tukkee.

**Data curation:** Ahmed Sahib Tukkee.

**Formal analysis:** Ahmed Sahib Tukkee, Noor Izzri bin Abdul Wahab.

**Funding acquisition:** Ahmed Sahib Tukkee.

**Investigation:** Ahmed Sahib Tukkee, Noor Izzri bin Abdul Wahab.

**Methodology:** Ahmed Sahib Tukkee, Noor Izzri bin Abdul Wahab.

**Project administration:** Ahmed Sahib Tukkee.

**Resources:** Ahmed Sahib Tukkee, Noor Izzri bin Abdul Wahab.

**Software:** Ahmed Sahib Tukkee.

**Supervision:** Noor Izzri bin Abdul Wahab, Nashiren Farzilah binti Mailah, Mohd Khair Bin Hassan.

**Validation:** Ahmed Sahib Tukkee, Noor Izzri bin Abdul Wahab, Nashiren Farzilah binti Mailah, Mohd Khair Bin Hassan.

**Visualization:** Noor Izzri bin Abdul Wahab.

**Writing – original draft:** Ahmed Sahib Tukkee, Noor Izzri bin Abdul Wahab.

**Writing – review & editing:** Ahmed Sahib Tukkee, Noor Izzri bin Abdul Wahab, Nashiren Farzilah binti Mailah, Mohd Khair Bin Hassan.

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
