## [Decision Letter · Decision Letter 0]

21 Nov 2023

PONE-D-23-29832Optimal performance of stand-alone hybrid microgrid systems based on integrated techno-economic-environmental energy management strategy using the gray wolf optimizerPLOS ONE

Dear Dr. Tukkee,

Thank you for submitting your manuscript to PLOS ONE. After careful consideration, we feel that it has merit but does not fully meet PLOS ONE’s publication criteria as it currently stands. Therefore, we invite you to submit a revised version of the manuscript that addresses the points raised during the review process.

We look forward to receiving your revised manuscript.

Kind regards,

Manish Kumar Singla

Academic Editor

PLOS ONE

4. We note that Figure 1 in your submission contain copyrighted images. All PLOS content is published under the Creative Commons Attribution License (CC BY 4.0), which means that the manuscript, images, and Supporting Information files will be freely available online, and any third party is permitted to access, download, copy, distribute, and use these materials in any way, even commercially, with proper attribution. For more information, see our copyright guidelines: http://journals.plos.org/plosone/s/licenses-and-copyright.

Reviewers' comments:

Reviewer's Responses to Questions

**Comments to the Author**

1. Is the manuscript technically sound, and do the data support the conclusions?

Reviewer #1: Yes

2. Has the statistical analysis been performed appropriately and rigorously? 

Reviewer #1: Yes

3. Have the authors made all data underlying the findings in their manuscript fully available?

Reviewer #1: Yes

4. Is the manuscript presented in an intelligible fashion and written in standard English?

Reviewer #1: Yes

5. Review Comments to the Author

Reviewer #1: It is a well written paper on using an optimizer (GWO in this case) in a hybrid microgrid. Analysis and results presentation are detailed and showing improvement of KPI. Authors also show a comparative results of using similar optimizer such as PSO and FA. Literature review was details.

I have no reservation to recommend the manuscript for acceptance.

6. PLOS authors have the option to publish the peer review history of their article (what does this mean?). If published, this will include your full peer review and any attached files.

Reviewer #1: No

---

## [Author Response · Author response to Decision Letter 0]

31 Dec 2023

I would like to express my gratitude to you and the reviewers for taking the time to evaluate my manuscript . I appreciate the constructive feedback and insightful comments provided during the review process.

I have carefully considered each of the reviewers' comments and suggestions, and I am pleased to inform you that I have made the necessary revisions to address their concerns. I believe these changes have significantly strengthened the quality and clarity of the manuscript.

In response to Reviewer 1's comments regarding [Figure 1 in our submission contain copyrighted images], I have removed the copyrighted images from Figure 1 and replaced them with illustrations prepared by the authors. Additionally, I have implemented all the magazine’s requirements mentioned in the revisions report. I believe these revisions have enhanced the overall coherence and impact of the paper.

---

## [Editor Report · Decision Letter 1]

18 Jan 2024

Optimal performance of stand-alone hybrid microgrid systems based on integrated techno-economic-environmental energy management strategy using the grey wolf optimizer

PONE-D-23-29832R1

Dear Dr. Tukkee,

We’re pleased to inform you that your manuscript has been judged scientifically suitable for publication and will be formally accepted for publication once it meets all outstanding technical requirements.

Kind regards,

Manish Kumar Singla

Academic Editor

PLOS ONE
---

## [Editor Report · Acceptance letter]

29 Jan 2024

PONE-D-23-29832R1 

PLOS ONE

Dear Dr. Tukkee, 

I'm pleased to inform you that your manuscript has been deemed suitable for publication in PLOS ONE. Congratulations! Your manuscript is now being handed over to our production team.

Kind regards, 

on behalf of

Dr. Manish Kumar Singla 

Academic Editor

PLOS ONE